# How Does Cultivar Affect Sugar Profile, Crude Fiber, Macro- and Micronutrients, Total Phenolic Content, and Antioxidant Activity on *Ficus carica* Leaves?

Candela Teruel-Andreu [1], Esther Sendra [1], Francisca Hernández [1] and Marina Cano-Lamadrid [2,*]

1 Centro de Investigación e Innovación Agroalimentaria y Agroambiental (CIAGRO-UMH), Miguel Hernández University, Ctra. Beniel, Km 3.2, 03312 Orihuela, Alicante, Spain
2 Postharvest and Refrigeration Group, Department of Agronomical Engineering and Institute of Plant Bio-Technology, Universidad Politécnica de Cartagena, 30203 Cartagena, Murcia, Spain
* Correspondence: marina.cano@upct.es; Tel.: +34-966749608

**Abstract:** The objective of this research was to evaluate the effect of the cultivar on the nutritional and functional parameters of *Ficus carica* leaves. This information will provide the basis for their potential use and future incorporation in other food matrices as food ingredients. Sucrose, glucose, and fructose were detected in all fig leaves, with mean values of 48.94, 66.74, and 43.70 g kg$^{-1}$ dried weight (dw), respectively. The crude fiber range was between 6.53% and 22.67%, being an interesting source of fiber. The most abundant macronutrient was calcium (Ca), followed by potassium (K) and magnesium (Mg). All cultivars showed high concentrations of iron (Fe). *Ficus carica* leaves can be a good material for obtaining extracts rich in fiber and calcium and provide an alternative source of these compounds to be incorporated into other nutraceutical and/or food matrices.

**Keywords:** fig; bioactive compounds; functionality; mineral content





## 1. Introduction

Today, there is an interest in searching for new sources of natural antioxidants due to the demand for healthier and "closer to zero waste" products [1]. Therefore, recent research has focused on the study of the nutritional composition and biological properties of different nonedible parts of plants, which can be called biomass waste. This biomass is rich in plant secondary metabolites, including bioactive compounds that play a key role in the defense against physiological and environmental stimulators, and the adaptation of plants to their environment. These compounds have extensive applicability in human health [2]. New sources of ingredients that are primary metabolites (sugars, fiber, etc.) are also of interest. All mentioned extracts/compounds may be used to increase the functional and techno-functional properties of new food matrices. They also can be of use in the pharmaceutical industry. According to Karim et al. (2012) [3], there are at least 14 groups of plant secondary metabolites with nutraceutical potential distributed in different anatomical parts of plants. For this reason, many authors are focusing their research on the study of leaves, for example, the leaves of *Citrullus colocynthis* [4], *Anredera cordifolia* leaves [2], *Arbutus unedo* leaves [1], *Mulberry leaves* [5], *Moringa olifeira* leaves [6,7], and other fruit tree leaves such as apple, pear, quince, apricot, peach, plum, sour cherry and sweet cherry [8].

The fig tree (*Ficus carica* L.; *F. carica*) is the most well-known *Ficus* species plant in the *Moraceae* family and is native to the Sub-Himalayan region and central India, although it is widely farmed around the world. *F. carica* is a species that has been widely farmed for its fruit and nutritional values [9]. The leaves are stipulated and petiolated with obovate, nearly orbiculate or ovate leaf blade, palmately lobed, cordate base, undulate or irregularly dentate margin, acute to obtuse apex, and scabrous-pubescent surfaces [10]. The therapeutic properties of *F. carica* have been used in traditional medicine practices such as Ayurveda,

Unani, and Siddha [11]. Fruits, roots, and leaves are used in traditional medicine to treat various conditions [12]. Recently, *F. carica* has been included in occidental Pharmacopoeias (i.e., Spanish Pharmacopoeia, British Pharmacopoeia) and therapeutic guides based on herbal medicines, such as the Physician's Desk Reference for Herbal Medicines (2000) [13]. Some recent research suggested the anticancer activities of *F. carica* leaf extracts [14] and another study found positive effects of the extract of *F. carica* leaves on liver cancer and colon cancer [9]. Also, another study showed that the ethanol extract of *Ficus carica* leaves promotes cancer cell death [15]. Other studies investigated and evaluated *F. carica* leaf extracts and found promising biological activities, such as hepatoprotective activity [10,11], the hypoglycemic effect [11], hypocholesterolemic activity of the decoction of leaves [11,16], hypolipidemic activity [10,11], strong antimicrobial activities [10,11], free radical scavenging activity [11], anti-HSV effect [10,11] and immunostimulant properties [11].

The modern pharmaceutical and food industry considers biomass waste as an almost infinite resource for functional product development [10]. For this reason, many authors are focusing their research on the analysis of *F. carica* leaves. A recent review compiled at least 40 bioactive compounds in *F. carica* leaves [17]. Some studies demonstrated that the phenol content in *F. carica* leaves is higher than that in either red wine or tea [18]. Some studies have reported the phenolic profile of fig leaves, which is composed of seven phenolic compounds, namely 3-CQA [3-*O*-caffeoylquinic acid], 5-CQA [5-*O*-caffeoylquinic acid], Q-3-Glu [quercetin 3-*O*-glucoside], Q-3-rut [quercetin 3-*O*-rutinoside], ferulic acid, psoralen, and bergapten. Other works have shown that in *F. carica* leaves, rutin, umbelliferone, and psoralen were the most abundant flavonoids, followed by coumarin and furanocoumarin compounds [18]. On the other hand, the presence of volatile compounds mainly distributed in *F. carica* leaves were alcohols, ketones, esters, sesquiterpenes, and norisoprenoid [13]. Therefore, due to the health-promoting potential of these compounds, the valorization of leaves with sustainable technologies to recover these high-value-added ingredients and their utilization in novel food formulation developments should be further investigated [19]. Until now, different potential uses of *F. carica* leaves have been reported and compiled in a recent review such as their use in nanoparticles, as antibacterial extracts, and as an additive for pasteurized milk for increasing shelf-life, among others [17].

For all the above-mentioned reasons, the objective of this research was to evaluate the effect of the cultivar on the nutritional and functional parameters of *Ficus carica* leaves. This information will provide the basis for their potential use as additives or extracts by the food and pharmaceutical industries. This is the first study that compares and characterizes the nutritional and functional parameters of the leaves of the *Ficus carica* of four dark varieties (the most relevant from the commercial point of view of southeastern Spain).

## 2. Materials and Methods

### 2.1. Vegetal Material

In this study, the leaves of four biferous varieties: San Antonio (SA), Colar ©, Cuello Dama Negra (CDN), and Superfig (SF) of *Ficus carica* were collected from the experimental field of the Universidad Miguel Hernández de Elche (UMH) in the province of Alicante Spain (02°03′50″ E, 38°03′50″ N). The Colar variety was collected in two zones: (i) in the above-mentioned coordinates (Colar UMH, CUMH) and, (ii) in a commercial plot in Albatera, Alicante, southern (CA) Spain (0°55′49″ W, 38°13′17″ N). The leaves were collected from 20-year-old trees. Fig trees were trained to a vase-shaped system and planted at a spacing of 8 m × 5 m. They were drip irrigated and were subjected to the standard farming practices (pruning, thinning, fertilization, and pest control treatments). Thirty leaves were collected one week after the fruit had been collected in May and July 2021, respectively. Leaves were moved to the laboratory for processing.

### 2.2. Leaf Characterization

Leaf characterization was done following The International Plant Genetic Resources Institute guidelines [20]. A sample size of 30 adult leaves per variety (10 random leaves per

tree) was characterized. All of them were taken from all tree orientations and middle parts of shoots, and only healthy and undamaged ones were selected. The parameters assessed were leaf length (from the base of the petiole to the tip of the central lobe, expressed in cm), leaf width, petiole length, and length of the central lobe (Figure 1).

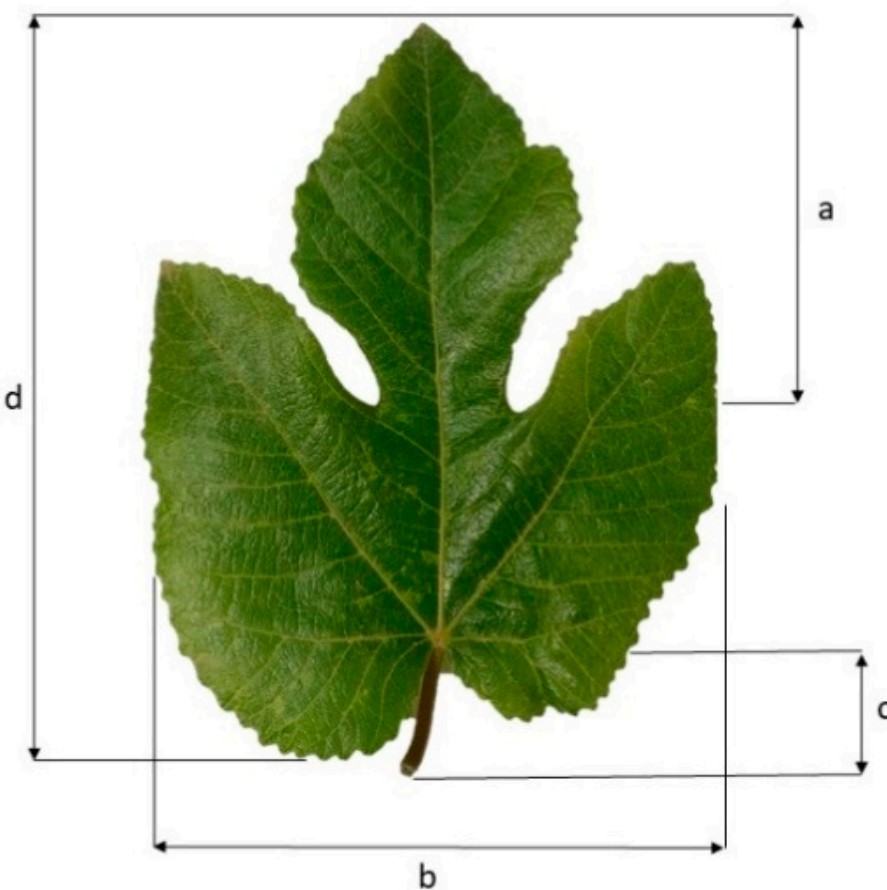

**Figure 1.** Representation of the measures considered to determine the size of the leaves, where "a" represents length of central lobe (cm); "b" leaf width (cm); "c" petiole length (cm); and "d" leaf length (cm).

### 2.3. Sugar Profile

Sugars were identified and quantified as previously described by Cano-Lamadrid et al. [21], with some modifications. Freeze-dried samples finely ground (0.2 g) were homogenized with 5 mL of phosphate buffer 50 mM (pH = 7.8) and centrifuged at 15,000× $g$ for 15 min at 4 °C (Sigma 3–18 K; Osterode and Harz, Germany), Then, 1mL of supernatant was filtered through a 0.45 µm Millipore filter (Billerica, MA, USA) before HPLC analysis. Then, 10 µL were injected into a Hewlett-Packard high-performance liquid chromatography (HPLC) series 1100 (Hewlett-Packard, Wilmington, DE, USA). The elution system consisted of 0.1% phosphoric acid with a flow rate of 0.5 mL min$^{-1}$. The sugars were eluted through a Supelco column [SupelcogelTM C-610H column (30 cm × 7.8 mm)] coupled with a Supelguard column (5 cm × 4.6 mm, Supelco, Inc., Bellefonte, PA, USA) and detected with a refractive index detector (Hewlett-Packard, series 1100, G1362A, Wilmington, DE, USA). Analyses were run in triplicate, and results were expressed as g kg$^{-1}$ dry weight (dw).

### 2.4. Crude Fiber, Macro- and MicroNutrient Content

Crude fiber content was determined according to methodology established by the Spanish Ministry of Agriculture, Fisheries, and Food as previously described by Sánchez et al. [22] using an ANKOM200/220 fiber analyzer (ANKOM Technology, Macedon, NY, USA). Each sample was tested in triplicate, and results were to be expressed as % dw. To determine the mineral content, 0.2 g of freeze-dried leaves were digested in a microwave (MARS

ONE, 240/50 CEM) reaching 200 °C in 15 min and held at this temperature for 15 min after the addition of 10 mL of concentrated, 65% ($w/v$), $HNO_3$. Later, samples were filtered with quantitative filter paper and transferred to a volumetric flask, and dilutions 1:10, 1:20, and 1:60 in the case of potassium were prepared using ultrapure deionised water, 18 MΩ (Milli-Q® system; Millipore Corporation, Madrid, Spain). Determination of macronutrients (Ca, Mg, and K) and micronutrients (Cu, Fe, Mn, and Zn) in previously mineralised samples was performed using an (ICPMS-2030, Shimadzu). Each sample was tested in triplicate, and results were to be expressed as g kg$^{-1}$ dw for macroelements (Ca, Mg, K, and Na) and as mg kg$^{-1}$ dw for microelements (Fe, Cu, Mn, and Zn).

### 2.5. Antioxidant Activity (AA) and Total Polyphenols Content (TPC)

Freeze-dried samples (leaves) (0.5 g) were blended with 10 mL of MeOH/water (80:20, $v/v$) + 1% HCl, sonicated at 20 °C for 15 min, and left at rest for 24 h at 4 °C. Later, the extract was again sonicated for 15 min and centrifuged at 15,000 rpm for 10 min. This extract was used to measure antioxidant activity by three methods (ABTS, DPPH, and FRAP methods) and total polyphenolic content (TPC). The radical scavenging activity was assessed using the DPPH radical (2,2-diphenyl-1-picrylhydrazyl) method, as described by Brand-Williams et al. (1999) [23], while the ABTS [2,2-azinobis-(3-ethylbenzothiazoline-6-sulfonic acid)] radical cation and ferric reducing antioxidant power (FRAP) methods were measured as previously described by Re et al. (1995) [24], and Benzie et al., (1996) [25], respectively. All analyses were done using a UV-visible spectrophotometer (Helios Gamma model, UVG 1002E, UK). A calibration curve (3.5–5.0 mmol Trolox L$^{-1}$) with good linearity ($R^2 \geq 0.999$) was used for the quantification. Analyses were performed in triplicate and results were expressed as mM Trolox dw. Total polyphenols content (TPC) was quantified using Folin–Ciocalteu reagent as described by Singleton et al. (1965) [26]. Absorption was measured using a UV–visible spectrophotometer (Helios Gamma model UVG 1002E, Helios, Cambridge, UK). All determinations were performed in triplicate and results were expressed as grams of gallic acid equivalent (GAE) per kilogram of dw.

### 2.6. Statistical Analyses

Data were analyzed using StatGraphics Plus, version 5.0 (Manugis-tics, Inc., Rockville, MD, USA). A one-way analysis of variance (cultivar as factors) was performed and mean values were compared by Tukey's multiple range test. Also, a one-way analysis of variance (location as a factor) was performed, and mean values were compared by Tukey's multiple range test.

## 3. Results and Discussion
### 3.1. Leaf Characterization

Table 1 shows the morphometrics parameters of the studied *F. carica* leaves. As to size, significant differences were found among cultivars, being CA cultivar, presented the highest value of leaf length, leaf width, length of the central lobe, and petiole length (27.46, 21.78, 14.55, and 10.54 cm, respectively). Taking the values of the morphometrics parameters of the rest of the cultivars into account, the range of leaf length, leaf width, length of central lobe, and petiole length was 19.65–22.71 cm, 17.38–21.04 cm, 10.16–12.64 cm, and 8.16–8.98 cm, respectively. Our results agreed with other authors that the size depends on the cultivar. Abdelsalam et al. (2019) [27] obtained differences in the leaf sizes of several *F. carica* cultivars, the smallest leaf length and leaf width size was 'Komesrey-El-Hammam' (5.4 and 6 cm), and the greatest leaf length and width values were found for 'Abodey-Giza' (23.5 cm) and Black_Mission (23.0 cm), respectively. Almajali et al. (2012) [28] showed leaf length values of 'Kortomanee', 'Byadee', 'Kdaree', 'Ajlounee', and wild fig cultivars (25.78, 20.95, 18.80 cm, 15.3 cm, and, 14.6 cm, respectively). While, other authors [29] obtained a lower ranged value in the case of leaf lengths of 6.20–13.80 cm, leaf widths of 4.10–15.30 cm, and petiole lengths of 1.68–5.80 cm. For leaf length, the CA variety obtained higher values than those found in the literature, while for leaf width the data obtained in this study

are within the range of those found in the literature. These differences may be due to the different genotypes and to environmental factors. Many studies have shown that water stress reduces leaf area. On the other hand, when location factor was considered, significant differences were found for leaf morphological characteristics between CA and CUMH. The CA cultivar presented a larger leaf length, leaf width, length of the central lobe, and petiole length size than the CUMH cultivar. Another study [6] using mulberry leaves, collected in Orihuela, the same city as the UMH plot is located, where some fig trees used in this study are located, found that the mean values for leaf length and leaf width were lower than the mean values obtained in this study (1.66 and 2.02-fold lower), respectively. According to the results obtained by these authors and our results, cultivar, and crop location affect leaf size.

**Table 1.** Mean values of morphological characteristics of leaves of different varieties of *Ficus carica*.

| Variety [a] | Size (cm) | | | |
|---|---|---|---|---|
| | Leaf Length | Leaf Width | Length of Central Lobe | Petiole Length |
| | ANOVA Test [b] | | | |
| | *** | *** | *** | *** |
| | Tukey's Multiple Range Test [c] | | | |
| SA | $21.70 \pm 0.35$bc | $21.04 \pm 0.71$ab | $10.94 \pm 0.30$c | $8.16 \pm 0.29$b |
| CA | $27.46 \pm 0.56$aA | $21.78 \pm 0.39$aA | $14.55 \pm 0.35$aA | $10.54 \pm 0.39$aA |
| CUMH | $19.65 \pm 0.52$dB | $17.38 \pm 0.67$cB | $10.16 \pm 0.26$cB | $8.64 \pm 0.35$bB |
| CDN | $22.71 \pm 0.42$b | $19.09 \pm 0.27$bc | $12.64 \pm 0.27$b | $8.98 \pm 0.32$b |
| SF | $20.77 \pm 0.36$cd | $18.60 \pm 0.30$c | $10.52 \pm 0.28$c | $8.71 \pm 0.23$b |

[a] SA: San Antonio; CA: Colar Albatera; CUMH: Colar UMH; CDN: Cuello Dama Negra; SF: Superfig. [b] NS not significant at $p > 0.05$ and *** significant at $p < 0.001$, respectively. [c] Values (mean $\pm$ standard error; $n = 30$) followed by the same letter, within the same column, were not significantly different ($p > 0.05$), according to Tukey's least significant difference test. Lowercase letter shows significant differences among cultivars, and capital letter shows significant differences between location of the same cultivar.

*3.2. Sugar Profile and Crude Fiber Content*

Table 2 shows the sugars identified and quantified in leaves (fructose, glucose, and sucrose). Our results indicated that fructose was the major sugar in leaves (1.10 times higher than glucose and 1.84 times higher than sucrose) (calculations were made with the average of all the varieties). However, other authors reported that sucrose was the highest sugar found in *F. carica* leaves and that the total sugar concentration ranged from 10.7% (Fracasana cv.) to 20% (Kalamon cv.) [30]. Among cultivars, significant differences were found, being the highest value of sugars in the CA cultivar (159.38 g kg$^{-1}$ dw). It should be clear that the same cultivar (Colar) between two locations (Albatera and UMH) showed differences, being attributed to the influence of agronomic factors such as differences in the irrigation of water (the UMH plot is irrigated with river water while the Albatera plot is irrigated with well water with a higher salt content), different location elevation (already mentioned in the plot description), and fertilization (no fertiliser was applied on the UMH plot while potash was used as fertilizer on the Albatera plot). The CA cultivar showed a 95.96, 14.77- and 3.55-times higher content than CUMH for sucrose, glucose, and fructose, respectively. It is essential to mention that the rest of the cultivars were in the same location with the same irrigation water and soil type, among other environmental conditions. Considering the values of total sugar content found in the *F. carica* leaves, these leaves could be used as natural sweeteners instead of other synthetic sweeteners.

As for crude fiber results, significant differences were observed between cultivars, being in the range between 6.53% and 22.67% for CUMH and CA, respectively. The author's hypothesis is the effect of water salinity on the content of pectin and fiber as mentioned above as the reason for the detected significant differences in the crude fiber content between both locations. These values are slightly lower than those reported by El Dessouky

Abdel-Aziz et al. [31]. The results showed that *F. carica* leaves can be a good source of fiber, especially if these results are compared with the crude fiber content of the fig fruit (2.2%) reported in another study (in this study Sultani fig trees' cultivar was used, the differences may be due to the cultivar) [32], being in agreement with results previously reported by Rusmadi et al. (2020) [33] for *F. carica* fruits of a different cultivar (0.88, 2.58, and 3.36%) [34]. It is important to mention that our results are expressed as crude fiber, this being necessary for future research to focus on total soluble and insoluble dietary fiber. The addition of fig leaf powders in other food matrices could be a good strategy to enhance fiber content to reach fiber contents suited to the mentions of 'high in fiber' or 'source of fiber'. In order to claim that a food is 'high in fiber' a minimum content of 6 g of fiber per 100 g of food is needed [35].

**Table 2.** Sugars profile [g Kg$^{-1}$ dry weight (dw)] and crude fiber content (%) of different varieties of *Ficus carica* leaves.

| Variety [a] | Sucrose | Glucose | Fructose | Crude Fiber |
|---|---|---|---|---|
| | | ANOVA Test [b] | | |
| | *** | *** | ** | ** |
| | | Tukey´s Multiple Range Test [c] | | |
| SA | 0.47 ± 0.01b | 4.47 ± 0.09b | 12.43 ± 0.29b | 16.75 ± 0.37a |
| CA | 48.94 ± 5.74aA | 66.74 ± 9.24aA | 43.70 ± 10.02aA | 22.67 ± 3.35aA |
| CUMH | 0.51 ± 0.01bB | 4.52 ± 0.03bB | 12.32 ± 0.10bB | 6.53 ± 0.35bB |
| CDN | 0.52 ± 0.00b | 4.61 ± 0.03b | 12.53 ± 0.08b | 19.3 ± 0.62a |
| SF | 0.51 ± 0.01b | 4.61 ± 0.04b | 12.62 ± 0.11b | 20.67 ± 0.1a |

[a] SA: San Antonio; CA: Colar Albatera; CUMH: Colar UMH; CDN: Cuello Dama Negra; SF: Superfig. [b] NS not significant at $p > 0.05$; ** and *** significant at $p < 0.01$ and 0.001, respectively. [c] Values (mean ± standard error; $n = 30$) followed by the same letter, within the same column, were not significantly different ($p > 0.05$), according to Tukey's least significant difference test. Lowercase letter shows significant differences among cultivars, and capital letter shows significant differences between location of the same cultivar.

### 3.3. Mineral Content

Table 3 shows the mineral composition of the leaves of the five *F. carica* cultivars studied. Significant differences ($p < 0.05$) were found in the content of macro- and microminerals between all the varieties studied. The leaves of the SA cultivar obtained the highest contents for all the macro- and microelements except for Mn, which was the CA cultivar that showed the highest content. Leaves from all cultivars had a high content of calcium (Ca), ranging from 19.97 g kg$^{-1}$ dw (CUMH) to 68.04 g kg$^{-1}$ dw (SA), followed by potassium (K) ranging from 13.87 g kg$^{-1}$ dw (CDN) to 18.63 g kg$^{-1}$ dw (SA). A large variation was observed in the contents of magnesium (Mg) 2.57–8.46 g kg$^{-1}$ dw and sodium (Na) 0.27–1.64 g kg$^{-1}$ dw. Among the microminerals, the contents of copper (Cu) varied from 4.18–13.11 mg kg$^{-1}$ dw; iron (Fe) varied from 201.07–342.21 mg kg$^{-1}$ dw; manganese (Mn) varied from 33.07–66.56 mg kg$^{-1}$ dw; and zinc (Zn) varied from 15.03–55.90 mg kg$^{-1}$ dw. For macro- and microelement contents, taking the location factor into account, the only significant differences were found between CA and CUMH cultivars in Ca, Mg, Na, and Mn, being the CA cultivar, in which the highest content was found. There were few studies on the mineral content in *F. carica* leaves depending on the cultivar. Therefore, the results of this research could increase the knowledge of the effect of cultivar on the mineral content of leaves of the *Ficus carica* in the same soil conditions. Our results are in agreement with previous studies conducted by other researchers [31,36,37], in which calcium is found as the main mineral in the leaves of fig trees [31,36]. The values obtained by other authors were lower values than ours, being in the range of 13.98–15.70 g kg$^{-1}$ dw. Calcium is a macroelement that adds a nutrient essential to the body's metabolism, and calcium deficiency is linked to osteoporosis [38]. The Codex Alimentarius, Guidelines for Use of Nutrition Claims states that solid foods must contain a calcium content of 15% of the nutrient reference value (NRV) of 800 mg of calcium to be labelled as a source of

calcium [39]. Therefore, the *Ficus carica* leaves supplementation of food matrices could be a good strategy to be able to use a nutritional claim. Regarding microelements, iron (Fe) showed a higher content of all microelements for leaves of the different cultivars tested. The differences found between the micronutrient contents in the different varieties may be due to the plants' tendency to accumulate greater amounts of micronutrients when there is a greater vegetative growth, therefore the lower the vegetative growth, the lower the concentration of elements [40].

**Table 3.** Macro (Ca, K, Mg, and Na; g $Kg^{-1}$ dw) and microelements (Cu, Fe, Mn, and Zn; mg $Kg^{-1}$ dw) of *Ficus carica* leaf.

| | Macroelements | | | | Microelements | | | |
|---|---|---|---|---|---|---|---|---|
| **Variety** [a] | **Ca** | **K** | **Mg** | **Na** | **Cu** | **Fe** | **Mn** | **Zn** |
| | ANOVA Test [b] | | | | | | | |
| | *** | * | *** | *** | *** | ** | *** | *** |
| | Tukey´s Multiple Range Test [c] | | | | | | | |
| SA | 68.04 ± 1.37a | 18.63 ± 1.10a | 8.46 ± 0.51a | 1.64 ± 0.08a | 13.11 ± 0.77a | 342.21 ± 26.16a | 60.13 ± 3.83a | 55.90 ± 5.33a |
| CA | 28.32 ± 0.30bA | 17.24 ± 0.72abA | 3.17 ± 0.07cA | 0.32 ± 0.01bA | 4.18 ± 0.29bA | 226.35 ± 15.10bA | 66.56 ± 3.76aA | 15.23 ± 1.22bA |
| CUMH | 19.97 ± 1.06cB | 15.03 ± 0.72bA | 2.57 ± 0.09dB | 0.27 ± 0.00cB | 4.73 ± 0.04bA | 201.07 ± 11.25bA | 33.07 ± 1.36bB | 18.73 ± 1.63bA |
| CDN | 23.87 ± 0.46bc | 13.87 ± 0.18c | 4.58 ± 0.08b | 0.40 ± 0.03b | 4.44 ± 0.68b | 225.37 ± 3.13b | 37.91 ± 0.37b | 15.03 ± 2.58b |
| SF | 26.72 ± 0.12b | 18.41 ± 3.12a | 3.03 ± 0.16c | 0.43 ± 0.02b | 5.80 ± 0.07b | 275.21 ± 9.21ab | 58.56 ± 1.68a | 22.29 ± 0.42b |

[a] SA: San Antonio; CA: Colar Albatera; CUMH: Colar UMH; CDN: Cuello Dama Negra; SF: Superfig. [b] NS not significant at $p > 0.05$; *, **, and ***, significant at $p < 0.05$, 0.01, and 0.001, respectively. [c] Values (mean ± standard error; $n = 30$) followed by the same letter, within the same column, were not significantly different ($p > 0.05$), according to Tukey's least significant difference. Lowercase letter shows significant differences among cultivars, and capital letter shows significant differences between location of the same cultivar.

*3.4. Antioxidant Capacity and Total Phenolic Content*

Table 4 shows the antioxidant capacity by three method assays and the total phenolic content by a Folin assay. The radical-scavenging activity by the ABTS assay of fig leaves' cultivars revealed the highest antioxidant activity was shown by cultivar SF (52.43 mM Trolox dw) followed by fig cultivars CDN (52.07 mM Trolox dw), SA (44.91 mM Trolox dw), CUMH (42.46 mM Trolox dw), and CA (33.81 mM Trolox dw). On the other hand, significant differences were found only between the CA and CUMH cultivars for the ABTS method. The DPPH assay of fig leaves' cultivars showed the highest activity to fig cultivar SF (72.45 mM Trolox dw), followed by fig cultivars CUMH (70.14 mM Trolox dw), CA (68.84 mM Trolox dw), CDN (59.27 mM Trolox dw), and SA (52.54 mM Trolox dw respectively). Regarding FRAP, the order of cultivars from highest to lowest was CDN (124.79 mM Trolox dw) > SF (115.66 mM Trolox dw) > CUMH (67.15 mM Trolox dw) > CA (60.70 mM Trolox dw) > SA (56.09 mM Trolox dw). It can be observed that the highest content of the antioxidant activity, measured by the three methods (sum of ABTS, DPPH, and FRAP methods) was for the SF cultivar, followed by CUMH, while the lowest antioxidant activity was for the SA cultivar. As to TPC, the highest TPC for cultivar CUMH (18.86 g GAE $kg^{-1}$ dw), followed by SA (18.62 g GAE $kg^{-1}$ dw), CDN (18.06 g GAE $kg^{-1}$ dw), SF (17.83 g GAE $kg^{-1}$ dw), and CA (16.64 g GAE $kg^{-1}$ dw). The main phenolic compounds found in the leaves of the *Ficus carica* were compiled previously by Teruel-Andreu et al. (2021) [17], being phenolic acids (caffeoylmalic acid, 3-*O*-caffeoylquinic acid, 5-*O*-caffeoylquinic acid) and flavonols (quercentin and kaempherol derivatives), among others. Caftaric acid was the highest value reported in *Ficus carica* leaves. Although more studies about the individual phenolics that were found in the different Ficus carica leaves, the antioxidant activity can be correlated with these mentioned compounds. Mahmoudi et al. [41] also found that the total phenolics content varies depending on the varieties, detecting higher total phenolics content for the biferous "Dhokkar" variety followed by the uniferous "Hamra" (46.074 mg GAE $g^{-1}$ dw and 42.889 mg GAE $g^{-1}$ dw, respectively). Several studies have shown that the content of total phenols in fig leaves is influenced by the type of solvent used for extraction. Authors included in previous studies comparisons of different solvents since there are few scientific manuscripts related to this topic, specifically with the same solvent used. Thus, Ghazi et al. (2016) [36] indicated that TPC, DPPH, and FRAP values

were higher in the methanolic extract of *Ficus carica* leaf (412.37 mg GAE 100 g$^{-1}$, 63.29 and 131.39 mmol Fe$^{2+}$ 100 g$^{-1}$, respectively), in comparison with the total phenolic content of water and methanol extracts. However. Gillani et al. (2012) [42] showed that the TPC of the water extract was highest as compared to the TPC of the methanol extract. Accordingly, extractions for antioxidant capacity and total phenolic content analysis in this study have been made with methanol extracts. On the other hand, other authors [43] studied the content of bioactive compounds with antioxidant capacity in the leaves of different fruit trees and found a high variation with results of α-tocopherol equivalents in a range of 74.14 (μg g$^{-1}$ dw) in sweet cherry to 194.22 (μg g$^{-1}$ dw) in apricot. They also found differences in the content of bioactive compounds with an antioxidant capacity between the leaves collected at two weeks after blooming and leaves collected at two weeks after fruits had been collected. Leaves collected two weeks after blooming had the highest contents of bioactive compounds with antioxidant activity. Thus, in the present study, the leaves were collected at the same time to see the effect of the cultivar only.

**Table 4.** Antioxidant activity (ABTS, DPPH, and FRAP; mM Trolox dw) and total polyphenol content (TPC; g GAE kg$^{-1}$ dw) of *Ficus carica* leaves.

| Variety [a] | ABTS | DPPH | FRAP | TPC |
|---|---|---|---|---|
| | | ANOVA Test [b] | | |
| | *** | *** | *** | * |
| | | Tukey´s Multiple Range Test [c] | | |
| SA | 44.91 ± 0.26b | 52.54 ± 1.48b | 56.09 ± 2.58c | 18.62 ± 0.64a |
| CA | 33.81 ± 1.08cB | 68.84 ± 0.41aA | 60.70 ± 1.70bcA | 16.64 ± 0.57bA |
| CUMH | 42.46 ± 1.03bA | 70.14 ± 2.33aA | 67.15 ± 1.98bA | 18.86 ± 1.05aA |
| CDN | 52.07 ± 0.11a | 59.27 ± 1.67b | 124.79 ± 1.14a | 18.06 ± 0.22a |
| SF | 52.43 ± 0.53a | 72.45 ± 1.05a | 115.66 ± 1.44a | 17.83 ± 0.15ab |

[a] SA: San Antonio; CA: Colar Albatera; CUMH: Colar UMH; CDN: Cuello Dama Negra; SF: Superfig. [b] NS not significant at $p > 0.05$; *and *** significant at $p < 0.05$ and 0.001, respectively. [c] Values (mean ± standard error; $n = 30$) followed by the same letter, within the same column, were not significantly different ($p > 0.05$), according to Tukey's least significant difference test. Lowercase letter shows significant differences among cultivars, and capital letter shows significant differences between location of the same cultivar.

### 3.5. PCA Analysis

For a better understanding of the relationships among the sixteen statistically significant variables studied for varieties of *F. carica* leaves, a PCA was carried out (linear dimensionality reduction method for processing of multivariate data) (Figure 2). This statistical test was run for all varieties studied. Figure 2 shows the first two components of the correspondence analyses PCA plot, which explained 80.51% of the variability in the data. The PCA explained analytical variables in two axes, F1 46.80% and F2 33.71%. The results showed three groups can be differentiated based on the position of the samples along the F1-axis, as can be seen (Figure 2). The first group included the CA cultivar which was linked with all the variables related to the sugar content (sucrose, glucose, and fructose). A second group consisted of the SA cultivar which was related to all mineral content variables. Finally, the rest of the cultivars CUMH, CDN, and SF appear grouped together with antioxidant capacity variables.

**Biplot (F1 and F2: 80.51 %)**

**Figure 2.** Differences between cultivars for all variables analysed in this study (PCA). Legend: ● Study varible ● Cultivar.

## 4. Conclusions

It is essential to highlight that this study is the first to study morphometric parameters, sugar profile, crude fiber, mineral concentration, and antioxidant activity in fig tree leaves grown in Southeastern Spain. Among the analyzed leaf cultivars, significant differences were found in all studied parameters. Also, both locations of the Colar cultivar affected the studied parameters. In conclusion, it could be said that Colar de Albatera showed the highest sugar content, being a potential sweetener. San Antonio cultivar presented the highest content of macro and mineral elements, being a suitable raw material to enrich another food matrix. Superfig and Cuello de Dama Negra cultivars presented the highest antioxidant capacity (ABTS and DPPH for Superfig; and FRAP for Cuello de Dama Negra). Future research should be carried out to know the specific bioactive compound which presents antioxidant capacity. Due to their functional compounds and properties, fig tree leaves could be used by the food industry for their health benefits and for pharmaceutical purposes. In addition, their use could offer benefits to farmers ensuring the sustainable management of this waste. For future research, the incorporation of this material and/or its extracts on food matrices would be of great interest.

**Author Contributions:** Formal analysis and Writing, C.T.-A.; Conceptualization, supervision, F.H.; Conceptualization and Methodology, M.C.-L.; Visualization, supervision, E.S. All authors have read and agreed to the published version of the manuscript.

**Funding:** Project AICO/2021/326 financed by the Autonomous Community of the Comunidad Valenciana through Conselleria de Innovación, Universidades, Ciencia y Sociedad Digital.

**Institutional Review Board Statement:** Not applicable.

**Informed Consent Statement:** Not applicable.

**Data Availability Statement:** Not applicable.

**Conflicts of Interest:** The authors declare no conflict of interest.

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
