# Peer review of "How Does Cultivar Affect Sugar Profile, Crude Fiber, Macro- and Micronutrients, Total Phenolic Content, and Antioxidant Activity on Ficus carica Leaves?"

_agronomy, doi:10.3390/agronomy13010030_

Round 1

Reviewer 1 Report

Currently, it is important to look for alternatives for using all parts of plants due to the health benefits. For this reason, the topic of this paper is interesting. To improve the writing and quality of the paper, the following there are some observations:

General article

After reading the article, the information does not allow an understanding of how the cultivars affect the nutritional composition and antioxidant activity of the leaves, since it only describes changes in these quality parameters, so it is necessary to analyze the results in greater depth.

The title indicates "a Potential Food Supplement", and the article is mentioned nothing about the topic. Is information would be expected on types of matrices in which they could be incorporated? or conditions of this extract to be able to be included in different types of food matrices.

Introduction.

In the introduction are presented a number of associated investigations with the topic of the paper and even applications in the food industry, which does not allow us to determine what is the contribution of this research. So It is necessary to improve the description of the novelty of this research since it is not clear the knowledge gap must be filled

The aim of the introduction is different from the aim in the abstract. Check

Why fiber, minerals, and sugar are considered chemical parameters and not nutritional parameters?

Results.

Line 179. why there is talk of environmental changes, if the samples were all taken from the same place

Line 184. How variety can be a factor if precisely the morphometric characteristics of different varieties are being compared.

Line 185. those changes in growing conditions, how does it influence the quality of the leaves

Line 193. Check the calculations, the fructose content is 10 times more than sucrose and 4 times more than glucose, this relationship is not clear.

Line 200. It is important to indicate well what were those changes in the growing conditions that could lead to these significant changes since the increase is almost 5 times.

Line 206. I suggest changing the way to express the ranges, first the smallest value and then the largest. Revise all data ranges presented in the article.

Line 207. It would be important to indicate what the difference may be due to.

Line 252. This expression would be obvious since no other changes were made

Table 3. Review the statistical analysis, because reviewing the data reported for the samples CA and CUMH in the parameters Mg, Na, Cu, and Zn, there seems to be no significant difference.

Line 261 to 270. Include a further analysis of the data because only what is shown in the table is described.

Line 283. Taking into account that the solvent influences the extraction and therefore the estimation of the TPC and AA, a comparison must be made with studies that have carried out extraction processes under these conditions.

Line 294. An analysis should have been included in relation to the same variety but from different provenance. since if it was included in the study it was for a reason.

Conclusion

Line 320 "leaves grown in similar conditions in Spain", In other words, the crops at the Miguel Hernández University of Elche (UMH) and Albatera, Alicante, southern Spain, were the same, then why were the differences presented in the Colar sample?

Author Response

  1. Currently, it is important to look for alternatives for using all parts of plants due to the health benefits. For this reason, the topic of this paper is interesting. To improve the writing and quality of the paper, the following there are some observations.

The authors thank the reviewer for the time spent in improving the paper. The changes were included in the updated document R1.

General article

  1. After reading the article, the information does not allow an understanding of how the cultivars affect the nutritional composition and antioxidant activity of the leaves, since it only describes changes in these quality parameters, so it is necessary to analyse the results in greater depth.

Authors agree with the reviewer and authors improved the discussion of the manuscript.

  1. The title indicates "a Potential Food Supplement", and the article is mentioned nothing about the topic. Is information would be expected on types of matrices in which they could be incorporated? or conditions of this extract to be able to be included in different types of food matrices.

Authors agree with the reviewer and “potential food supplement” was removed in the title. Although authors mention the potential application, the aim of this research is not that. Future studies related to the food application of the leaves will be carried out.

Introduction.

  1. In the introduction are presented several associated investigations with the topic of the paper and even applications in the food industry, which does not allow us to determine what is the contribution of this research. So, it is necessary to improve the description of the novelty of this research since it is not clear the knowledge gap must be filled.

Done as suggest. Authors wrote in the first version of the manuscript a few words related to the importance of this research but as reviewer indicated, the novelty was missed. Authors included a sentence explaining the novelty of this study. Please view lines 87-90.

  1. The aim of the introduction is different from the aim in the abstract.

Done as suggest. Please view lines 14-17 and lines 87-90.

  1. Why are fiber, minerals, and sugar considered chemical parameters and not nutritional parameters?

Done as suggest. Authors agree with this comment. Authors included “nutritional” instead of “chemical”. Please view line 85.

Results.

  1. Line 179. why there is talk of environmental changes, if the samples were all taken from the same place. Line 184. How variety can be a factor if precisely the morphometric characteristics of different varieties are being compared. Line 185. those changes in growing conditions, how does it influence the quality of the leaves.

Done as suggest. As reviewer suggested statistics were carried out to compare the same cultivar in different locations as vegetal material indicated (Lowercase letter shows significant differences among cultivars, and capital letter shows significant differences between location of the same cultivar in all Tables added in this manuscript). In this way, environmental/growing conditions are considering in all measured parameters. More information was added in the discussion of each parameter related to the location.

  1. Line 193. Check the calculations, the fructose content is 10 times more than sucrose and 4 times more than glucose, this relationship is not clear. Line 200. It is important to indicate well what were those changes in the growing conditions that could lead to these significant changes since the increase is almost 5 times.

Authors checked again the calculation and they were correct. This detected difference should be justified due to the different location and growing conditions. More information to justify these significant changes was added between lines 222 and 227.

  1. Line 206. I suggest changing the way to express the ranges, first the smallest value and then the largest. Revise all data ranges presented in the article.

Done as suggest. Please see lines:  233, 262-268, 338 and 339, among others. Authors checked the whole manuscript.

  1. Line 207. It would be important to indicate what the difference may be due to.

Done as suggest. Please view line 234-236.

  1. Line 252. This expression would be obvious since no other changes were made.

The reviewer is right it has been removed. Please view line 285-287.

  1. Table 3. Review the statistical analysis, because reviewing the data reported for the samples CA and CUMH in the parameters Mg, Na, Cu, and Zn, there seems to be no significant difference.

Data was checked and some modifications were carried out. In the text, extra information was added as reviewer suggested. errors have been corrected for Ca, K, Mg, Na and Cu. Please see lines 269-272.

  1. Line 261 to 270. Include a further analysis of the data because only what is shown in the table is described.

Done as suggest. Section 3.4 was improved.

  1. Line 283. Considering that the solvent influences the extraction and therefore the estimation of the TPC and AA, a comparison must be made with studies that have carried out extraction processes under these conditions.

The reviewer is right but as there is a few scientific manuscripts about this topic, authors decided to compare with the published evidence although the solvents used were different. Line 327 was added: “Authors included previous studies comparing with different solvents because there are few scientific manuscripts related to this topic, specifically with the same solvent used”

  1. Line 294. An analysis should have been included in relation to the same variety but from different provenance. since if it was included in the study it was for a reason.

The statistical comparison between both locations of the Colar cultivar has been included (Lowercase letter shows significant differences among cultivars, and capital letter shows significant differences between location of the same cultivar. Please see Tables and legend of each table.

Conclusion

  1. Line 320 "leaves grown in similar conditions in Spain", In other words, the crops at the Miguel Hernández University of Elche (UMH) and Albatera, Alicante, southern Spain, were the same, then why were the differences presented in the Colar sample?

Done as suggest.  As authors explained in above answers, the study is related to 4 cultivars and one of them was collected from two locations. It is true that both location is in southern Spain. Please view line 377-379.

Reviewer 2 Report

This manuscript is well-written and structured. However, I would invite the authors to consider the following comments to improve its overall quality.

1-      the rational is conducted but needs to be more clear

2-      please define any acronyms before using them

3-      please check the manuscript for some typos

Author Response

Reviewer 2

  1. This manuscript is well-written and structured. However, I would invite the authors to consider the following comments to improve its overall quality.

The authors thank the reviewer for the time spent in improving the paper. The changes were included in the updated document R1.

  1. The rational is conducted but needs to be more clear. Please define any acronyms before using them. Please check the manuscript for some tipos

The manuscript has been revised considering reviewer`s suggestions for instance: dried weight (dw) or Ficus carica (F. carica).

Reviewer 3 Report

The manuscript has investigated the influence of cultivar on the physicochemical properties of fig leaves. The manuscript is well-structured and interesting results are reported. It can be accepted after minor revisions.

Comments:

Line 36: the reference style is not correct.

Line 55: Do you mean leaf extract?

Line 260: Please explain why the fig leaves show antioxidant activity and which phenolic compounds are found in fig leaves.

Author Response

  1. The manuscript has investigated the influence of cultivar on the physicochemical properties of fig leaves. The manuscript is well-structured and interesting results are reported. It can be accepted after minor revisions.

The authors thank the reviewer for the time spent in improving the paper. The changes were included in the updated document R1.

  1. Line 36: the reference style is not correct.

Done as suggested. Authors check all the manuscript and changes were done.

  1. Line 55: Do you mean leaf extract?

Authors agree with the reviewer that some information was missed. Now, the authors changed the sentence. Please see line 58-60

  1. Line 260: Please explain why the fig leaves show antioxidant activity and which phenolic compounds are found in fig leaves.

Done as suggested. The section 3.4. was carefully checked and the required information was added.
